# Proteolysis of *β*-Lactoglobulin Assisted by High Hydrostatic Pressure Treatment for Development of Polysaccharides-Peptides Based Coatings and Films

Yang Fei [1,2], Zhennai Yang [1], Sobia Niazi [3], Gang Chen [1,2,4,*], Muhammad Adnan Nasir [5], Imran Mahmood Khan [3,*], Abdur Rehman [3], Rana Muhammad Aadil [6], Monica Trif [7,*] and Viorica Coşier [8]

1 Beijing Advanced Innovation Center for Food Nutrition and Human Health, Beijing Technology and Business University, Beijing 100048, China
2 College of Food and Health, Zhejiang Agriculture and Forest University, Hangzhou 311300, China
3 School of Food Science, Jiangnan University, Wuxi 214122, China
4 School of Food Science and Technology, Henan University of Technology, Zhengzhou 450001, China
5 Department of Allied Health Sciences, The University of Chenab, Gujrat 50700, Pakistan
6 National Institute of Food Science and Technology, University of Agriculture, Faisalabad 38000, Pakistan
7 Department of Food Research, Centre for Innovative Process Engineering (Centiv) GmbH, 28857 Syke, Germany
8 Faculty of Animal Science and Biotechnology, University of Agricultural Sciences and Veterinary Medicine Cluj-Napoca, 400372 Cluj-Napoca, Romania
* Correspondence: chenggangdeng@126.com (G.C.); imk2654@jiangnan.edu.cn (I.M.K.); mt@centiv.de (M.T.)

**Abstract:** Peptides usually have many bioactive functions. The variety of peptide binding and the modularity of the components allow for their application to additional tissues and materials; hence broadening the range of possible coatings and films. *β*-lactoglobulin (b-LG) forms spherical microgels or can be used in the formation of coated particles, with the core formed by aggregated b-LG and the coat by polysaccharides. The enzymatic proteolysis of b-LG assisted by high hydrostatic pressure (HHP) treatment was studied. Pretreatment of HHP enhanced the hydrolysis degree (DH) of b-LG. The highest value of DH without pretreatment was 24.81% at 400 MPa, which increased to 27.53% at 200 MPa with pretreatment, suggesting a difference in the DH of b-LG caused by the processing strategy of HHP. Molecular simulation suggested that the flexible regions of b-LG, e.g., Leu140-Ala142 and Asp33-Arg40, might contribute to enzymatic proteolysis. The b-LG hydrolysate exhibited the highest capacity of scavenging free DPPH and OH radicals at 200 MPa. In addition, the 1–2 kDa and 500–1000 Da peptides fractions significantly increased from 10.53% and 9.78% (under 0.1 MPa) to 12.37% and 14.95% under 200 MPa, respectively. The higher yield of short peptides under HHP contributed to the antioxidant capacity of b-LG hydrolysates. Enzymatic hydrolysis also largely reduced the immunoreactivity of b-LG, which is of high importance in the practical application of b-LG in the field of coatings and films in regard to biocompatibility. Hydrolysis of b-LG assisted by high-pressure treatment showed promising potential in the preparation of bioactive peptides for further development of polysaccharide-peptide-based coatings and films.

**Keywords:** high hydrostatic pressure; *β*-lactoglobulin; enzymatic hydrolysis; polysaccharides-peptides

## 1. Introduction

Peptides derived from food proteins are considered effective therapeutic food ingredients for people who are unable to digest dietary proteins [1] and are used as natural health-care compounds added to food, pharmaceuticals, medicine, and cosmetics [2], or in the development of peptides-based antimicrobial coatings and films. The variety of peptide binding and the modularity of the components allow for their application to additional tissues and materials, hence broadening the range of the coating and films [3]. There are three major ways to include peptides: directly into the polymer; as a coating on the

polymeric surface; and by immobilizing them within the polymer. *β*-lactoglobulin (b-LG) forms spherical microgels or can be used in the formation of coated particles (with a protein core and a polysaccharide shell), the core being formed by aggregated b-LG and the coat by polysaccharides (alginate, k-carrageenan, pectin, etc.) [4–7]. The *β*-LG and various anionic polysaccharides can be heated to a temperature greater than the protein's denaturation temperature to produce stable polymer particles. Food protein-derived peptides can also exert bioactive function in addition to their basic nutritional roles [8]. Enzymatic hydrolysis has been one of the main ways to produce bioactive peptides from different foods [9]. In enzymatic proteolysis, the cleavage site of proteins is crucial for producing protein peptides, which is associated with the functionality of the peptides [10]. Physical modifications, such as thermal treatment and pressure treatment, enable exposure or burying of the potential cleavage sites of protein molecules by altering protein conformation [11]. High hydrostatic pressure (HHP) is an emerging technique in food processing and food preservation [12,13], which has the capacity of altering the hydrolysis efficiency of enzymes on proteins [14]. Pressure treatment at 200 MPa was shown to contribute to a high yield of peptides from *α*-casein [15]. In addition, the bioactive properties of obtained peptides may be improved in this way. For example, the anti-inflammatory effect and antioxidant activity of phosvitin have been improved by HHP [16]. However, to date, it is still unclear how hydrolysis under HHP contributes to more effective proteolysis [17].

Beta-lactoglobulin (b-LG), the primary protein in whey, has been a desirable resource of bioactive peptides [18]. Hydrolysis of b-LG has the capacity to improve not only the protein bioavailability but also to increase allergenicity safety [19]. Changes in the b-LG structure during or after pressure treatment have been used to deduce the conformational states consisting of the swollen state and the molten globule state [20]. In addition, the threshold for b-LG unfolding at pH 3 under HHP was found to be around 300 MPa [21].

The conformational changes of b-LG may cause the exposure of some special amino acid residues, such as the free thiol group of Cys121, and Trp19 [22]. Additionally, previous studies reported a higher rising impact of HHP on enzymes under HHP [18]. The peptide profiles in all hydrolysates were expected to be essentially similar to the hydrolysates of native b-LG and b-LG subjected to HHP [23]. In general, HHP treatment of proteins results in high efficiency of proteolysis by various enzymes, which may be attributed to the exposure of cleavage sites of the proteins for the sequential interaction with enzymes [24].

HHP treatment of b-LG allows the steering of the enzymatic hydrolysis influencing the release of peptides from proteins. Thus, in this work, we have studied the hydrolysis of b-LG subjected to HHP treatment (Figure 1) and evaluated the bioactive properties of released peptides. In addition, molecular dynamics simulation was used to elucidate the molecular mechanism. This work will provide important clues and elucidation for HHP application to assist enzymatic hydrolysis of proteins as promising candidates in the preparation of bioactive peptides for further development of polysaccharides-bioactive peptides-based antimicrobial coatings and films. However, it should be noted that polysaccharides, because of their free carboxyl groups, can gel with polyvalent cations such as calcium, barium, and strontium ions, as well as polypeptides. Ion affinities for their units are not identical, and, thus, further research is needed.

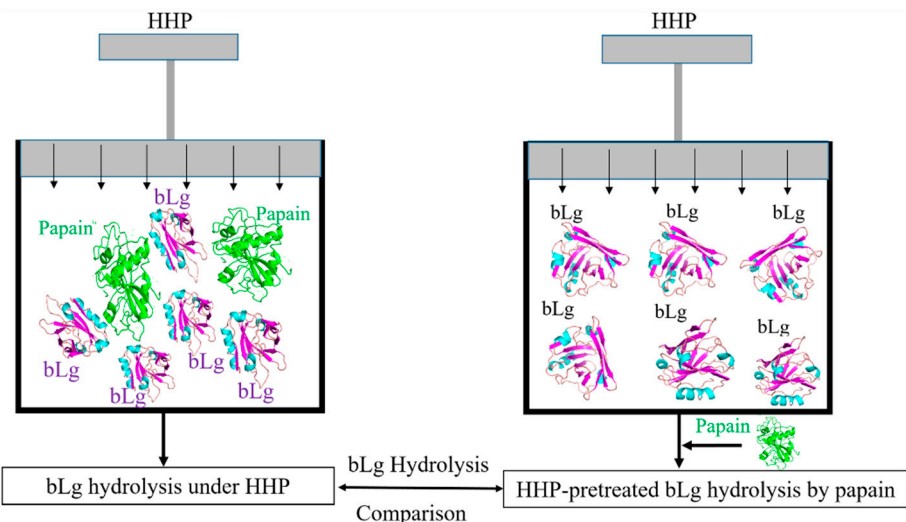

**Figure 1.** Schematic comparison of b-LG (b-LG) hydrolysis for different pressure treatments.

## 2. Materials and Methods

### 2.1. Materials

The b-LG (98%), 1,1-diphenyl-2-trinitrophenylhydrazin (DPPH), and serine used in this study were supplied by Sigma-Aldrich (Shanghai, China). Bromelain (14 U/mg), papain (≥2000 U/mg), and trypsin (≥180 U/mg) were bought from Sangon Inc. (Shanghai, China). The o-phthaldialdehyde (OPA) α-deoxyribose, 2-thiobarbituric acid, and dithiothreitol (DTT) were supplied by Solarbio Science and Technology Co., Ltd. (Beijing, China). All other chemical reagents were of analytic grade.

### 2.2. Enzymatic Hydrolysis

The preparation of a 1 mg/mL b-LG solution was performed using Tris-HCl buffer of 25 mM (pH 7.0) and hydrolysis for 0–120 min using papain, trypsin, and bromelain, respectively. The proteolysis reaction was performed at 40 °C under atmospheric pressure unless otherwise stated. The hydrolysis reaction was ended by inactivating the enzymes by heat treatment at 90 °C for 10 min. Treated reaction solutions were preserved at −80 °C for analysis.

### 2.3. HHP and Thermal Treatment of the Papain

The papain solution was properly prepared using Tris-HCl buffer (pH 7.0, 25 mM). For determining the effect of temperature on papain, the activity of papain was determined at temperatures within the range of 30 to 70 °C with casein as the substrate. The activity of papain was determined using the method described by Homaei and Samari [25]. A 0.1 mg/mL enzyme solution was diluted with a volume ratio of 1:7. Subsequently, a 400 μL solution containing 0.45 mM casein was mixed with an equal volume of diluted enzyme solution and incubated for 10 min at different temperatures. The reaction was ended by adding 800 μL of 10% trichloroacetic acid aqueous solution. The absorbance of the mixtures was recorded at 280 nm. The 1.0 mg/mL papain solution was loaded into the chamber using a compressible polypropylene tube and treated by various pressures from 0.1 to 600 MPa for 30 min. The activities of all treated samples were rapidly measured at 60 °C. The relative activity was presented as the ratio of the activity of the treated papain to that of native papain determined at 60 °C.

### 2.4. HHP Treatment on b-LG

HHP treatment was performed using a MICRO FOODLAB high-pressure device (Standard Fluid Power Ltd., Harlow, UK). Preheating was carried out to obtain a designed temperature of the pressure chamber before HHP treatment. The b-LG was dissolved in

Tris-HCl buffer (25 mM, pH 7.0). The protein solution and enzyme solution were mixed and filled in a 5 mL polypropylene tube with a screw lid, followed by HHP treatment under different pressures from 0.1 to 600 MPa for the hydrolysis under HHP for 30 min. The rate for pressurization was fixed at 20 MPa/s. After pressure processing, all reactions were terminated by 90 °C heating for 10 min. For comparison, pretreatments of b-LG by pressure were carried out under pressures from 0.1 to 600 MPa for 30 min. Afterward, the pressurized b-LG was hydrolyzed by papain under 0.1 MPa at 60 °C for 30 min before determination.

### 2.5. Degree of Hydrolysis Determined Using the OPA Reaction

The degree of hydrolysis (DH) of b-LG was investigated according to the method of Nielsen et al. [26], which is based on the reaction of OPA with primary amino nitrogen, and detected by absorbance at 340 nm using a Shanghai Scientific spectrometer (Shanghai Scientific instrument Co., Ltd., Shanghai, China). The OPA solution was prepared by dissolving sodium tetraborate decahydrate, SDS, dithiothreitol, and OPA using deionized water. A 0.1 g/L standard serine solution was prepared using deionized water. In all determinations, 6-mL OPA solution was mixed with 800 μL of standard serine solution, the sample, and deionized water (blank), respectively. All measurements, carried out in triplicate, were reacted for 2 min at 25 °C. The degree of hydrolysis (DH%) was calculated from the absorbance values using the formula:

$$DH\% = h/h_{tot} \times 100\% \tag{1}$$

where $h_{tot}$ represents the total number of peptide bonds per protein equivalent substrate protein [meqv/g protein]; the value of 8.0 meqv/g protein was used for b-LG; h represents the free amino group amount in proteolysis and could be calculated according to Equation (2):

$$h = \frac{\text{Serine-NH2} - \beta}{\alpha} \times (\text{meqv/gprotein}) \tag{2}$$

where $\alpha$ and $\beta$ are 1.00 and 0.40, respectively; and Serine-NH$_2$ was the value of the serine solution at 340 nm.

The value of Serine-NH$_2$ was calculated using Equation (3):

$$\text{Serine-NH}_2 = \frac{A - A_c}{A_s - A_c} \times 0.9156 \times 0.1 \times \frac{100}{m \times w} \tag{3}$$

where $A_c$ and $A_s$ are the absorbance of the standard and the control, respectively, and m and w are the weight of the samples and the amount of protein in the samples (%), respectively.

### 2.6. DPPH Radical Scavenging Assay (%)

The radical scavenging ability of b-LG hydrolysates prepared under varying conditions was assessed using the method described by Yu et al. [27]. A 0.2 mg/mL ascorbic acid solution was assigned as a control. The hydrolysates were diluted to 4 mL with deionized water. A 1-mL solution of 40 μM 2,2-diphenyl-1-picrylhydrazyl (DPPH) radical in ethanol was mixed with the 4 mL sample and then bathed in water at 37 °C in the dark for 15 min.

The radical scavenging activity was estimated using Equation (4):

$$\%\text{DPPH free radical scavenging} = (A_c - A_s)/A_c \times 100 \tag{4}$$

where $A_c$ and $A_s$ are the absorbance of the control and samples, respectively.

### 2.7. Hydroxyl Radical (●OH) Scavenging Activity Analysis

Iron generates the hydroxyl radical through the Fenton reaction with $H_2O_2$, based on which the ·OH scavenging capacity was evaluated through the method described by Hanasaki et al. [28]. The reaction mixture comprised 0.2 mL of treated b-LG sample, 1 mL of 6 mM $FeSO_4$, and 1.8 mL of deionized water. After gentle mixing, 1 mL of 6 mM $H_2O_2$

was used to trigger reactions and the mixture was incubated in the dark at 25 °C for 10 min. Subsequently, 1.0-mL salicylic acid of 6 mM was added and mixed sufficiently. After 30 min, all absorbances were evaluated at 510 nm. Distilled water was assigned as a control.

•OH scavenging activity (S%) was estimated according to Equation (5):

$$•OH \text{ scavenging activity} = \left(1 - \frac{A_1 - A_2}{A_0}\right) \times 100\% \tag{5}$$

where

$A_1$ represents the absorbance of the samples,

$A_2$ represents the absorbance of the solution in which the salicylic acid–water is replaced by the same amount of water, and

$A_0$ represents the absorbance of the control.

### 2.8. Iron Chelating Activity

The ferrous ion ($Fe^{2+}$) chelating ability of the enzyme-treated b-LG was measured using the method of Jeong et al. [29]. The reaction system comprised 1815 μL of deionized water, 450 μL of treated b-LG, and 45-μL of 2 mM $FeSO_4$, being stirred and kept for 30 min at 25 °C. Then, 90 μL of 5 mM ferrozine was added to the reaction system. The absorbance was measured at 562 nm with deionized water as a control.

The ferrous ion chelating capacity was calculated using Equation (6):

$$\text{Iron chelating capacity} = (B_c - B_s)/B_c \times 100 \tag{6}$$

where $B_c$ and $B_s$ are the absorbance of the control and samples, respectively.

### 2.9. In Silico Analysis

The native b-LG structure was taken from the Protein Data Bank (ID: 3NQ9). GROMACS software 5.0 was employed to perform in silico analysis with a G43a1 force field and SPC water. The steepest descents method was assigned for energy minimization. The system was equilibrated with NPT and NVT ensembles for 0.5 ns, which was followed by all-atom simulations of 400 ns under different conditions. The LINCS algorithm was assigned to constrain all bonds in the in silico analysis. In addition, Bredesen's method was adopted to couple with an external bath and the isotropic Parrinello-Rahman protocol was taken in the generation of desired pressures (0.1, 200, 400, and 600 MPa). Periodic boundary conditions were applied to all simulations at 60 °C, respectively. The PyMOL package 2.0 was employed to elaborate the captures.

### 2.10. Estimation of Peptide Molecular Weight (Mw) Distribution

The Mw distribution of b-LG hydrolysates was estimated according to the method of Irvine [30]. The b-LG hydrolysates were measured utilizing high-performance size exclusion chromatography (HPSEC) on a Waters 2790 Separations Model (Waters Corp. Milford, MA, USA) with a TSK G2000SWxL gel column (Tosoh Corp., Yamaguchi prefecture, Japan). The b-LG treated by enzymes under HHP was filtered with 0.22 μm filters and fractioned through the HPLC. The elution was performed with acetonitrile solution (acetonitrile: Milli-Q water: trifluoroacetic acid = 45:55:0.1, *v/v*), at a 0.5-mL/min flow rate, and all corresponding absorbances of the samples were recorded at 220 nm. The injection volume was 20 μL. Two identical portions of each sample were injected to perform the analysis. For standards, Trp (204 Da), GLV (287 Da), SGNIGFPGPK (1114 Da), insulin (5700 Da), and myoglobin (17,600 Da) were used in the Mw estimation.

The Mw of samples was calculated using Equation (7):

$$\text{LogMw} = 3.8045 - 0.2658 \text{ T} \tag{7}$$

where $M_W$ denotes $M_W$ (Da) and T is the elution time (min).

### 2.11. Enzyme-Linked Immunosorbent Assay (ELISA)

An ELISA was performed to evaluate the allergenic property of the treated b-LG using the method of Chen and Yang [17]. To estimate the amount of b-LG hydrolysates-IgE in the samples, each polystyrene microplate well was precoated by 50 µL of b-LG-specific IgE in phosphate buffer (20 mM, pH 7.5) and then washed three times using the buffer including 0.05% (*w/v*) Tween 20. All samples were added in triplicate to the precoated wells. All wells were subsequently blocked using 100 µL of horseradish peroxidase-conjugated rabbit b-LG-specific IgE, covered with strips, and incubated for 1 h at 37 °C. Following the reaction, all wells were immediately washed three times, followed by the addition of 100 µL of chromogen reagent. Then, an incubation after gentle mixing was carried out at 37 °C for 15 min. Similarly, the b-LG hydrolysates-IgG was determined using the above method with the IgE replaced with IgG from goat serum. After the chromogenic reaction was stopped, the amounts of b-LG hydrolysates-IgE and b-LG hydrolysates-IgG in each sample were determined by measuring the corresponding absorbance at 450 nm with a microplate reader (Thermo Fisher Scientific Co., Ltd., Waltham, MA, USA). All the samples were prepared and analyzed in triplicate.

### 3. Results

### 3.1. Enzymatic Hydrolysis of b-LG

The b-LG hydrolysis curves against three distinct proteases are shown in Figure 2 at an enzyme: substrate ratio of 1:25. The evolution of each hydrolysis was characterized by a fast-initial reaction rate (0–1 h), which was also followed by a relatively slow phase of a decreased hydrolysis rate within two hours. The highest degree of hydrolysis (DH) value of 20.57% was observed after hydrolysis by papain for 2 h (Figure 2); while the highest DH with trypsin and bromelain was 14.60% and 8.52%, respectively. It has been reported that the DH of native b-LG was 13.2% after hydrolysis by trypsin for 2 h [24].

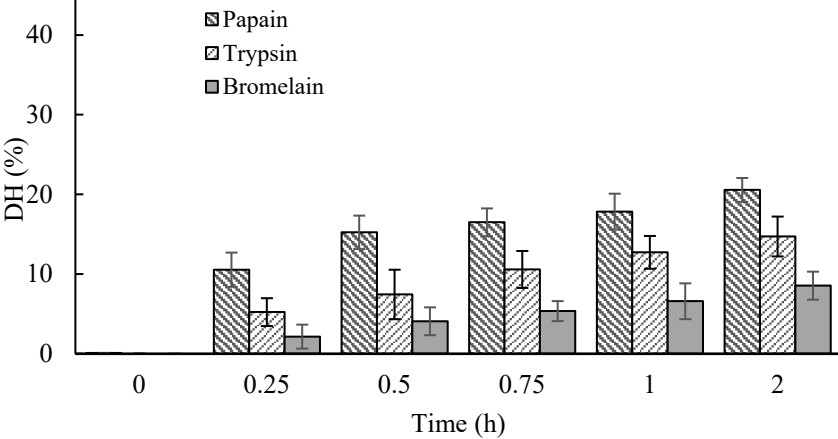

**Figure 2.** Hydrolysis of b-LG by papain, trypsin, and bromelain under 0.1 MPa at 40 °C.

The cleaving sites for hydrolyzing proteins varied according to the types of proteolytic enzyme [31]. Thus, the molecular profiles of hydrolysates were estimated after withdrawing samples following 15 min of reaction time. The results indicated the maximum Mw of up to 11.50 kDa of b-LG hydrolyzed by papain, while the trypsin partially hydrolyzed the b-LG with the estimated maximum Mw of 15.35 kDa which accounted for 67% of the peak area. Bromelain, on the other hand, demonstrated low proteolysis activity with the remaining b-LG, with an estimated maximum Mw of 19 kDa. Due to its high hydrolytic activity, papain was chosen as the model enzyme in subsequent experiments.

### 3.2. Effects of Temperature and HHP on Papain

To investigate the effect of temperature on papain's biocatalytic ability, the change in enzyme activity was estimated for treatments at temperatures ranging from 30 to 70 °C.

As shown in Figure 3a, the activity of papain under 0.1 MPa increased with temperature from 30 to 60 °C, reaching the highest value of 100%. The sequential decrease of the enzyme's activity at 70 °C is attributed to the thermal deactivation of papain [25]. The determined optimal temperature for papain proteolysis is consistent with the work of Smith and Hong-Shum [32], who reported that papain had an optimal temperature of 65 °C for its activity. According to Leeb et al. [24], the pre-denaturation of b-LG at 80 °C resulted in a decrease in b-LG DH and cleavage sites. As a result, in subsequent experiments, the reaction temperature was set at 60 °C.

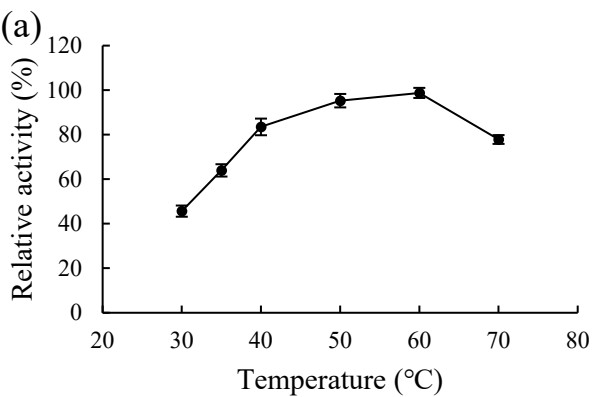
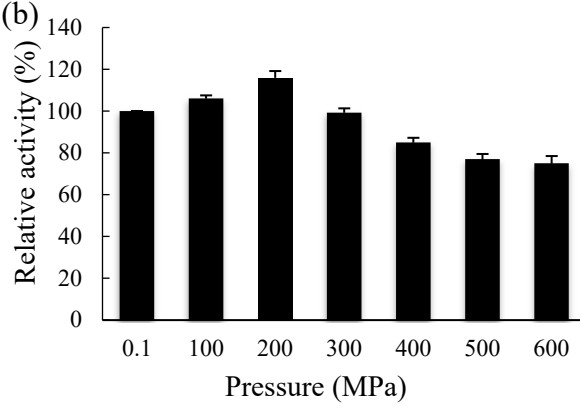

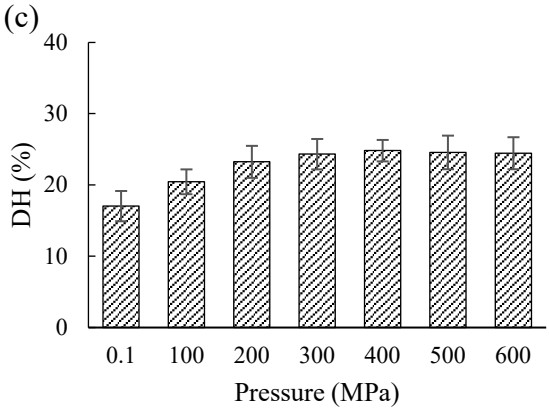
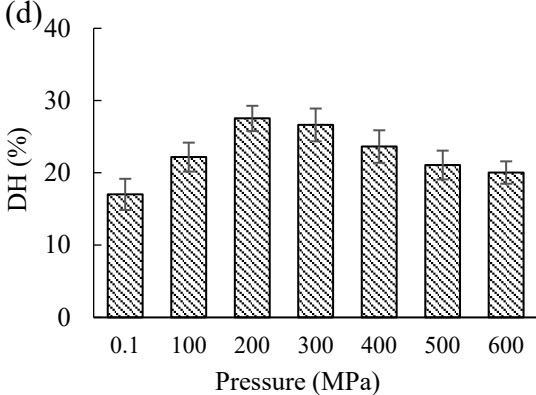

**Figure 3.** Effects of temperature and HHP on the activity of papain. (**a**,**b**) The DH of b-LG by papain in case of different high-pressure treatments, (**c**) The hydrolysis of b-LG at 60 °C after high-pressure pretreatment, and (**d**) The hydrolysis of b-LG under different high pressures at 60 °C.

The effects of HHP on papain were studied. As shown in Figure 3b, the increment in the pressure from ambient (0.1 MPa) to 200 MPa resulted in the enhancement of the relative activity of papain to 115.90%. The activity of papain then decreased as the pressure continuously increased up to 600 MPa. The data suggested that the increase in papain inactivity was highly dependent on the correct choice of pressure, which is consistent with previous work that reported papain inactivation at high pressure exceeding 400 MPa [33].

### 3.3. Effects of HHP on the DH of b-LG

The DH of pressure-treated b-LG samples is shown in Figure 3c. The DH of the b-LG samples after HHP treatment was higher than that of non-pressurized b-LG samples. In addition, the DH of b-LG remarkably increased by approximately 50% with the elevation of pressure up to 600 MPa, compared to the non-pressurized b-LG, presenting a slight variation in DH after the pressure processing at 400 MPa and 600 MPa. It has previously been reported that HHP-pretreating b-LG at a pressure of 200 MPa resulted in a higher DH

than that with the native b-LG [22]. The reason behind the higher DH at 200 MPa could be the alteration of protein cavities and the hydration shell of the b-LG molecule by changing the conformational arrangements of functional amino acid residues and increasing the susceptibility of the b-LG to proteolytic enzymes [34]. HHP also disrupts the molecular interactions stabilizing protein structure, e.g., hydrophobic interactions, promoting the hydrolysis of b-LG [35]. The high DH of HHP-pretreated b-LG should be attributed to the conformational changes of b-LG, which enhanced the enzyme accessibility of b-LG [36].

The hydrolysis of b-LG was carried out under higher pressure to investigate the DH of b-LG under HHP. The DH under HHP was found to be higher than that pretreated at 200 and 300 MPa. Enzymatic hydrolysis increased with the pressure level (0.1–300 MPa), with the maximum value of DH (27.53%) obtained at around 200 MPa, consistent with the work of Bamdad et al. [18], explaining the significant improvement in DH of b-LG by HHP treatment. The applied HHP treatment during proteolysis was more effective than pressure pretreatment before proteolysis. A similar result has also been documented with the higher efficiency of enzymes hydrolyzing other proteins under HHP conditions, such as the DH of bovine serum albumin and mushroom foot protein under 100–500 MPa [14,37]. However, as shown in Figure 3d, the DH of b-LG began to decrease as the pressure increased from 400 MPa to 600 MPa, and fell to 19.03% at 600 MPa.

Figure 3 demonstrates the inactivation of papain with the increase of high hydrostatic pressure over 300 MPa. Thus, the decrease in DH is attributed to the inactivation of papain by HHP. It is supposed that enzymatic hydrolysis under HHP is caused by the exposure of emerging cleavage sites during protein unfolding as well as the promotion of substrate-enzyme interactions [22]. The improved DH of protein by papain under HHP conditions has also been documented on other proteins, for instance, buckwheat protein [38]. The highly flexible region of b-LG caused by HHP may contribute to the fast hydrolysis of b-LG [21]. As a result, the higher DH of b-LG under HHP should be associated with the influence of pressure on both b-LG and papain.

### 3.4. Structural Changes in the b-LG Molecule

To clarify the effects of the conformational states of the b-LG molecule, the b-LG structure generated with MD simulation was superposed with its native conformation (Figure 4). The native b-LG structure was altered at 60 °C. As shown in Figure 4, some well-organized helical structures gradually lost their initial conformation as the HHP increased from 0.1 MPa to 600 MPa. Some $\beta$-sheet structural elements were also partially broken. For example, together with the alternation of the orange-colored helix (Leu140-Ala142) into a random coil, the sheet structures (Thr97-Asn90, Cys66-Lys75) labeled with the green, were shortened at 600 MPa compared with the native structure. In addition, the Val81-Ile84 $\beta$-sheet completely disappeared, changing into a random-coil structure under 600 MPa. The results indicated that high pressure disrupted the "barrel" shape of b-LG. In addition, the loop of Asp33-Arg40 showed various displacements, suggesting the flexibility of the peptide fragment. The helix containing Leu140-Ala142 and the helix connected with Asp33-Arg40 were the most flexible regions under HHP [39]. Those two regions may simultaneously greatly contribute to the hydrolysis of b-LG.

### 3.5. Peptide Profile Determined by HPSEC

The Mw distribution of b-LG hydrolysates produced by papain under HHP for 30 min is shown in Table 1. In the table, the Mw distribution is classified into six groups, 5000–7000 Da, 3000–5000 Da, 2000–3000 Da, 1000–2000 Da, 500–1000 Da, and <500 Da. The b-LG hydrolysate under 0.1 MPa showed a similar $M_W$ distribution, compared to that under HHP (100–600 MPa). The data in Table 1 revealed no residual intact b-LG molecules after proteolysis. The largest Mw was observed at 6.03 kDa at 0.1 MPa. Compared to the b-LG hydrolysate under 0.1 MPa, b-LG hydrolysates under HHP displayed fewer fractions of Mw over 3 kDa (Table 1). The amounts of the released peptides exhibited in the 1–2 kDa and 300–1000 Da fractions were greater enhanced from 10.53% and 9.78% under 0.1 MPa to

12.37% and 14.95% under 200 MPa, respectively (Table 1). While the peptides in the 5–7 kDa and 3–5 kDa fractions decreased from 6.88% and 10.64% under 0.1 MPa to 0.64% and 2.92% under 200 MPa, respectively. Furthermore, as the HHP increased from 200 to 600 MPa, the Mw fractions in the 1–5 kDa range changed slightly. Although the content of the various Mw fractions changed with pressure level, no new Mw fractions were produced, which is consistent with previous work [23]. It has been documented that chickpea hydrolysates by papain under 300 MPa exhibited high antioxidant capacity with peptide fractions of <3 kDa having the highest content, and primarily ranging within 500–1000 Da [40]. The change of Mw fractions may influence the antioxidant property of the b-LG hydrolysates.

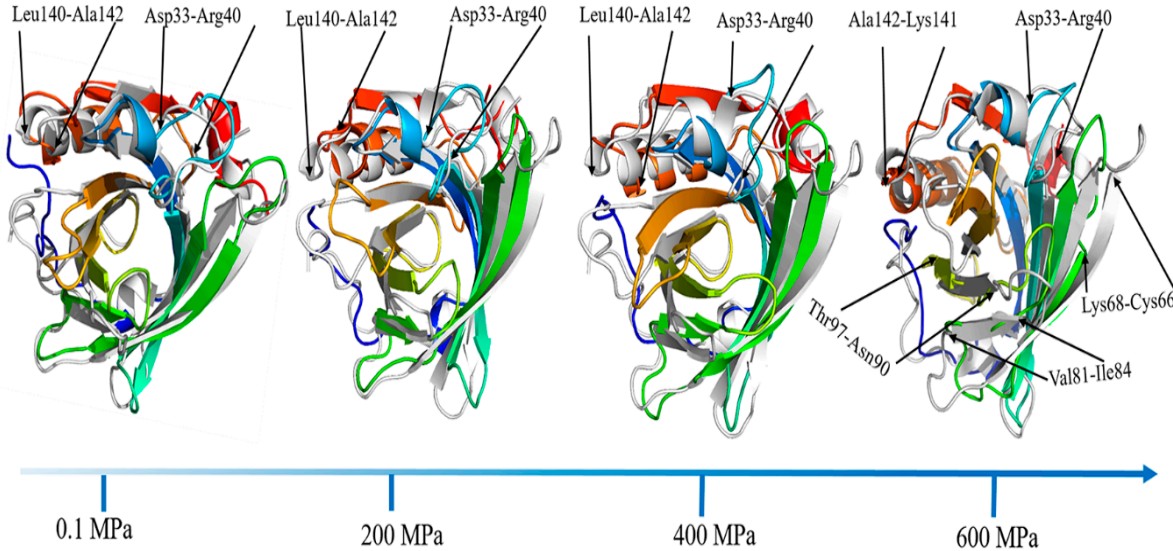

**Figure 4.** Conformational changes of b-LG under different pressures as revealed by molecular dynamics simulation. The structure colored gray is the initial structure of b-LG, while the rainbow-colored structures are the structure under different pressures.

**Table 1.** Molecular weight (Mw.) profile of b-LG hydrolysates under different pressures.

| Pressure (MPa) | Mw. Fraction (%) | | | | | |
|---|---|---|---|---|---|---|
| | 5–7 kDa | 3–5 kDa | 2–3 kDa | 1–2 kDa | 500–1000 Da | <500 Da |
| 0.1 | 6.88 | 10.64 | 5.56 | 10.53 | 9.78 | 56.62 |
| 100 | 2.67 | 7.11 | 4.68 | 10.29 | 9.76 | 65.49 |
| 200 | 0.64 | 2.92 | 4.8 | 12.37 | 14.95 | 64.31 |
| 300 | 0.83 | 3.87 | 5.58 | 12.8 | 13.60 | 63.38 |
| 400 | 0.86 | 4.19 | 5.80 | 12.78 | 13.82 | 62.55 |
| 500 | 0.94 | 5.29 | 4.99 | 11.41 | 11.62 | 65.75 |
| 600 | 1.84 | 6.96 | 5.23 | 12.02 | 11.75 | 62.20 |

*3.6. Antioxidant Capacity of b-LG Hydrolysates*

3.6.1. DPPH Scavenging Capacity

The radical scavenging capacity of DPPH is a simple index that can be used to characterize a compound's scavenging capacity [41]. Thus, the antioxidant capacity of the b-LG hydrolysates was estimated using DPPH free-radical scavenging activity and compared to that of ascorbic acid. As shown in Figure 5, b-LG hydrolysates produced under atmospheric pressure had higher DPPH scavenging activities than ascorbic acid.

The data in Figure 5a showed that all b-LG hydrolysates resulting from papain under HHP were capable of scavenging DPPH free radicals, with the highest scavenging activity of 73.63% being under 200 MPa. The b-LG hydrolysate at 300 MPa had a slightly lower scavenging activity of 72.2%. In addition, the scavenging activity of b-LG hydrolysates resulting from HHP showed higher values than ascorbic acid. The b-LG hydrolysates had a dose-

dependent DPPH scavenging capacity. Similarly, increased concentration of DH contributes to the increase of DPPH scavenging activity of other protein hydrolysates [18,42].

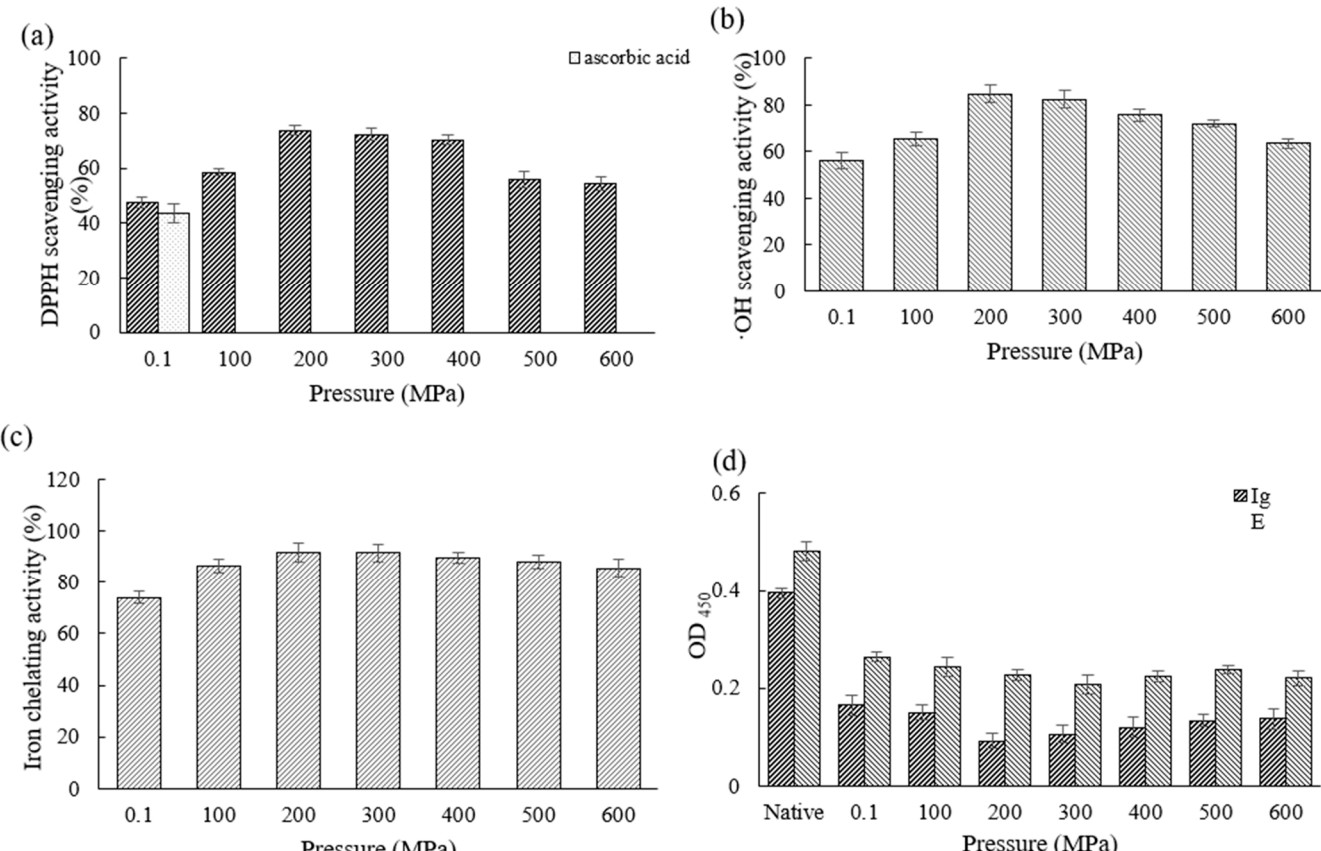

**Figure 5.** The antioxidant activities of hydrolysates produced under different pressures. (**a**) The DPPH scavenging activity of b-LG hydrolysates, (**b**) The ·OH scavenging activity of b-LG hydrolysates, (**c**) The ion chelating activity of b-LG hydrolysates, and (**d**) The OD value at 450 nm of IgE/IgG-b-LG hydrolysates after HHP treatment.

The antioxidant capacity does not necessarily increase as protein DH increases [43], as the antioxidant capacity of b-LG hydrolysates correlates with their specific amino acid sequence and amino acid composition [44]. It is proposed that a few amino acids, e.g., Trp, His, Lys, Tyr, and Met, possess antioxidant ability [45]. However, the peptides released from the b-LG by papain had no specific amino acid composition, as confirmed by previous reports. Thus, the result is mainly attributed to enhanced DH of b-LG as a result of HHP with an increased concentration of antioxidant peptides, which could stop the radical chain reaction by converting DPPH free radicals to stable chemicals.

### 3.6.2. Hydroxyl Radical (●OH) Scavenging Assay

The radical system used to evaluate antioxidant capacity may affect the test results, so more radical systems were needed to investigate the ability of given antioxidants to scavenge radicals. Thus, the ·OH scavenging capacities of b-LG hydrolysates were estimated. Figure 5b shows an obvious increment of the hydroxyl radical scavenging activity with pressure increasing from 0.1 to 200 MPa. Subsequently, it was followed by a gradual decrease in ·OH scavenging activity with the elevation of pressure from 300 MPa to 600 MPa. The hydroxyl radical scavenging activity was 87.98% under 200 MPa. Moreover, radical scavenging activity with HHP showed a similar evolution compared to the change of the DH of b-LG treated under high-pressure conditions. These results indicate that the high DH of b-LG resulting from the HHP treatment contributed to the change of hydroxyl

radical scavenging. Therefore, HHP mainly enhanced the antioxidant activities of b-LG hydrolysates by enhancing the concentration of functional peptides. HHP treatment has also been used to improve the anti-oxidative abilities of other protein hydrolysates. For example, SPPH hydrolyzed by the enzyme alcalase under high pressures (100–300 MPa) showed much higher scavenging activity on ●OH than those under 0.1 MPa [46]. Similarly, the increase in the amount of low-Mw hydrolysates was attributed to the improvement in the antioxidant ability of chickpea hydrolysates by HHP at 200 MPa [40]. The Mw distribution of b-LG hydrolysates also revealed an increase in the proportion of low Mw fractions. In addition, the DH was significantly improved by HHP. Thus, these two factors contributed to the increased scavenging activity of b-LG hydrolysate.

### 3.6.3. $Fe^{2+}$-Chelating Activity

In the ferrous ion-chelating test, $Fe^{2+}$ forms a complex with ferrozine leading to a red color, which allows us to analyze the chelating capacity of coexisting chelators by characterizing color changes [16]. As shown in Figure 5c, hydrolysis under high pressure largely improved the iron-chelating activity of the b-LG hydrolysates. The highest iron-chelating activity was 91.78% under 200 MPa, which is similar to that at 300 MPa (91.45%) and 400 MPa (89.53%). In addition, the chelating activity decreased as the pressure increased from 300 to 600 MPa. According to Bamdad et al. [18], the iron-chelating activity of b-LG hydrolyzed under 0.1 and 100 MPa showed a powerful iron chelating ability (at 0.5 mg/mL) in an amount-dependent manner. The results indicate that the HHP treatment is primarily responsible for the antioxidant activity of b-LG hydrolysate Ferrous ion chelation participates in ferrous ion chelation via coordination and electrostatic binding. These results suggest a significant increase of the exposed amino acids enabling the chelating of metal ions (i.e., cysteine and histidine) by enzymatic hydrolysis.

### 3.7. Influence of Enzymatic Hydrolysis under HHP on b-LG Immunoreactivity

Hydrolysis is an effective method for reducing or eliminating the allergenicity of protein allergens. To evaluate the effect of pyrolysis assisted by HHP on the immunoreactivity of the treated b-LG, an indirect in vitro ELISA assay was carried out. As Figure 5d shows, the OD of IgE-b-LG hydrolysates was reduced from 0.165 to 0.096 as the pressure increased from 0.1 to 200 MPa, achieving the lowest value around 200 MPa. While the OD value of IgE-b-LG hydrolysates increased with an elevation of pressure from 300 MPa to 500 MPa. The result showed a significant decrease in the IgE binding of b-LG compared to the IgE binding of non-hydrolyzed b-LG, which had a value of 0.395. The IgG-b-LG hydrolysates showed a similar trend with increasing pressure, and the lowest value of 0.207 was found at 200–400 MPa, indicating a lower IgG binding affinity after hydrolysis. In addition, the higher DH of b-LG corresponds to a lower OD value of IgE/IgG-b-LG, signifying that the hydrolysis contributed to the reduction of allergenicity of b-LG, in agreement with other reports about b-LG [47]. The retained residual immunoreactivity of b-LG also indicated a significant, but not complete, decrease in antigenicity caused by proteolysis of b-LG. The reason may be that peptides such as Ser21-Arg40, Val41-Lys60, and Leu149-Ile162, having IgE epitopes, were released, as confirmed in previous studies. However, the residual binding activity cannot cause antigenicity, indicating that hydrolysis is useful to reduce the immunoreactivity of b-LG.

To assess the biocompatibility of b-LG with various anionic polysaccharides, several in vitro and in vivo studies can be conducted to investigate the influence of the respective composition of the polysaccharides. The immunogenic response may be independent of the composition of polysaccharides and is more likely due to other natural compounds incorporated, such as proteins and polyphenolic compounds. Therefore, an effective reduction of the immunoreactivity of b-LG as performed in the present study is of high importance, before its use in practical coating applications of b-LG.

## 4. Conclusions

The DH of b-LG proteolysis under high hydrostatic pressure was higher than that with pretreatment by pressure. In addition, HHP treatment improved the antioxidant properties of b-LG hydrolysates, such as the capacity for scavenging free DPPH and ·OH radicals and the capacity of chelating $Fe^{2+}$. HHP treatment contributed to a higher yield of short peptides released in b-LG hydrolysates. The improved antioxidant properties of b-LG hydrolysates should be attributed to the higher concentration of short peptides released from b-LG by HHP. In addition, enzymatic hydrolysis effectively reduced the immunoreactivity of b-LG. A combination of enzymatic treatments and proper physical processing will provide a biotechnique for removing food allergenicity. At the same time, the use of antimicrobial peptides in the food industry is still a novel concept. The application of antimicrobial peptides to combat bacterial infectious agents in biofilms and coatings is still in its early stages, and the research on developing polysaccharides-peptides-based antimicrobial coatings and films is ongoing.

**Author Contributions:** Conceptualization, Y.F., G.C. and I.M.K.; methodology, Y.F., Z.Y. and S.N.; software, I.M.K. and M.A.N.; data curation, G.C., I.M.K., M.A.N. and A.R., writing—original draft preparation, Y.F.; writing—review and editing, G.C., I.M.K., M.A.N., A.R., R.M.A., V.C. and M.T. Supervision, G.C. and I.M.K. All authors have read and agreed to the published version of the manuscript.

**Funding:** This study was supported by the Beijing Advanced Innovation Center for Food Nutrition and Human Health (2018022), Beijing Technology and Business University (BTBU).

**Institutional Review Board Statement:** Not applicable.

**Informed Consent Statement:** Not applicable.

**Data Availability Statement:** All data that support the findings of this study are included within the article.

**Conflicts of Interest:** The authors declare that they have no known competing financial interest or personal relationships that could have appeared to influence the work reported in this paper.

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
