# Peer review of "Proteolysis of β-Lactoglobulin Assisted by High Hydrostatic Pressure Treatment for Development of Polysaccharides-Peptides Based Coatings and Films"

_coatings, doi:10.3390/coatings12101577_

Round 1

Reviewer 1 Report

In the current manuscript entitled “Proteolysis of β-lactoglobulin assisted by high hydrostatic pressure treatment releasing peptides and reducing immune reactivity for development of polysaccharides-peptides based coatings and films” Yang Fei and colleagues demonstrated the hydrolysis of β-lactoglobulin by papain under high hydrostatic pressure (HHP) treatment. The authors have shown the HHP treatment improved the antioxidative properties by scavenging free DPPH, OH radicals, and chelating of Fe2+. Overall, the current study found that the HHP treatment had a higher degree of hydrolysis than the pretreatment by pressure. The reasons might be attributed to conformational changes of bLg by loss of alpha-helices and β-sheets.

In my opinion, this report has enough novelty to meet the requirements for publication in Coatings after addressing the following comments:

1.    The title should be “Proteolysis of β-lactoglobulin assisted by high hydrostatic pressure treatment for development of polysaccharides-peptides based coatings and films”.

2. The abstract is lengthy and should be rewritten concisely.

3. Figure 1 is not precise and should be modified accordingly.

4. What is the difference between Fig 3c and 3d?

5. The elaboration of MD is missing, please include it.

6. What would be the reason for a substantial increment in the 5-7 k Da range from 500 to 600 in Table 1?

7. The manuscript is poorly written, please consider rewriting it wherever it is applicable.

Author Response

Feedback to Reviewer 1:

  1. The title should be “Proteolysis of β-lactoglobulin assisted by high hydrostatic pressure treatment for development of polysaccharides-peptides based coatings and films”.

Response: Thank you very much. Your comments on the manuscript are of great importance in improving our work. The title was simplified as suggested by the comments. (Page 1)

  1. The abstract is lengthy and should be rewritten concisely.

Response: Thank you very much. The abstract was shortened and rewritten. The title was simplified as suggested by the comments. (Page 1)

  1. Figure 1 is not precise and should be modified accordingly.

Response: Figure 1 has been revised accordingly. (Page 3)

  1. What is the difference between Fig 3c and 3d?

Response: Fig 3c was obtained with the hydrolysis of b-LG that was pretreated by high-pressure treatment. Fig 3c was obtained with the hydrolysis of bLg under high pressures. These two figures elucidated the difference in DH caused by the pressure processing way.

  1. The elaboration of MD is missing, please include it.

Response: The elaboration of MD was contained in the text. (Page 5)

  1. What would be the reason for a substantial increment in the 5-7 k Da range from 500 to 600 in Table 1?

Response: It is well known that papain is not a specific protein hydrolase and does not act on specific peptide bonds. The conformational changes of b-LG molecules and the pressure inactivation on papain all contributed to the variety of hydrolysis. The detailed changes in protein caused by pressure treatment have been reviewed by Silva et al. (High-pressure chemical biology and biotechnology. Chemical Reviews, 2014, 114 (14): 7239-7267.) Particularly, the degree and landscape of protein unfolding will be different under different pressures, which will expose some interaction sites for hydrolysis. These two aspects will mainly contribute to the mentioned observation.

  1. The manuscript is poorly written, please consider rewriting it wherever it is applicable.

Response: Thanks for the excellent observational and for highlighting these errors and ambiguities which we have corrected. We have gone through the entire manuscript carefully and adjusted every relevant sentence to avoid dangling modifiers and clarify our meaning.  We have corrected all the glaring and confusing errors and concise them for better understanding. We also seek professional help to improve the quality of the paper, modified the manuscripts, and also to improve or correct every ambiguity or inaccurate. In the revised manuscript, the English, grammatical, and format errors are corrected and the interpretations and flow have been significantly improved. The revised changes were marked in red text in the manuscript.

Reviewer 2 Report

The manuscript is scientific solid and very interesting. The data show that the hydrolysis of the β-lactoglobulin assisted by high hydrostatic pressure treatments is able to produce peptides with reducing immunoreactivity for development of polysaccharides-peptides based coatings and films.However, the effect of the temperature under kinetic energy of the catalytic steps must be used to compare the enzymes on different pressures. Also, the kinetic energy of the denaturation step must be determined.

Author Response

Feedback to Reviewer 2

  1. The manuscript is scientific solid and very interesting. The data show that the hydrolysis of the β-lactoglobulin assisted by high hydrostatic pressure treatments is able to produce peptides with reducing immunoreactivity for development of polysaccharides-peptides based coatings and films. - Response: Thank you for your appreciation.
  2. However, the effect of the temperature under kinetic energy of the catalytic steps must be used to compare the enzymes on different pressures. Also, the kinetic energy of the denaturation step must be determined. - Response: Thank you for your suggestions. In this study and at this point the enzymatic hydrolysis largely reduced the immunoreactivity of b-LG, which is it of high importance needed to be researched before a practical application of b-LG in the field of coatings and films when comes to biocompatibility.

Reviewer 3 Report

The authors have reported the bioactivity of β-lactoglobulin treated with high hydrostatic pressure. These results will be helpful and informative for researchers in the field of biomaterials and materials chemistry. Whereas the reviewer thinks that the authors’ study in this manuscript is quite interesting and suggestive, some descriptions are not enough. The authors’ manuscript is not suitable for publication in “Coatings” in the present form.

From these considerations, the reviewer recommends to accepting for publication in " Coatings," if the following issues are resolved.

(1)   Title, “Proteolysis of β-lactoglobulin assisted by high hydrostatic pressure treatment releasing peptides and reducing immuno-reactivity for development of polysaccharides-peptides based coatings and films”: The results and discussion of coatings and films should be required for publication in “Coatings.

(2)   Page 12, line 449, "In order to assess biocompatibility of b-LG with various anionic polysaccharides, several in vitro and in vivo studies were conducted to investigate an influence of the respective composition of the polysaccharides.": The results of this experiment should be presented. These results and a detailed discussion are the most important part of this study.

(3)   What kind of special polysaccharides did the authors use in this study?

Author Response

Reviewer 2:

(1) Title, “Proteolysis of β-lactoglobulin assisted by high hydrostatic pressure treatment releasing peptides and reducing immuno-reactivity for development of polysaccharides-peptides based coatings and films”: The results and discussion of coatings and films should be required for publication in “Coatings.”

Response:

We highly appreciated the reviewer's suggestion.

(2)   Page 12, line 449, "In order to assess biocompatibility of b-LG with various anionic polysaccharides, several in vitro and in vivo studies were conducted to investigate an influence of the respective composition of the polysaccharides.": The results of this experiment should be presented. These results and a detailed discussion are the most important part of this study.

Response:

We highly appreciated the reviewer's suggestion for a more exhaustive discussion to better express its practical application. However, the application of antimicrobial peptides to combat bacterial infectious agents in bio-films and coatings is still in its early stages, and the research on developing polysaccharides-peptides-based antimicrobial coatings and films is ongoing.

Before in vitro and in vivo studies, biocompatibility is an important step to be resolved for a practical application. Therefore the hydrolysis of b-LG assisted by by high pressure treatment showed promising potential in the preparation of bioactive peptides for further development of polysaccharides-peptides based coatings and films. Although, we agree with the reviewer’s suggestion and appreciated highlighting in vitro and in vivo studies to investigate the influence of the respective composition. We already working on these aspects but they are our future projects which still need a long time to accomplish. We have also done changes in the last paragraph of the discussion in view of future projects.

(3)   What kind of special polysaccharides did the authors use in this study?

Response: An effective reduction of the immunoreactivity of b-LG as performed in the present study is of high importance before a practical coating application of b-LG which needs to be fulfilled. Polysaccharides, because of their free carboxyl groups, can gel with polyvalent cations as well as polypeptides. Ion affinities for their units are not identical, and therefore this is a further step in research.

Round 2

Reviewer 3 Report

(1)   Title, “Proteolysis of β-lactoglobulin assisted by high hydrostatic pressure treatment for development of polysaccharides-peptides based coatings and films”: If the title says "coatings and films," a discussion of coatings and films should be provided in the main text of the authors’ manuscript.

(2)   “However, the application of antimicrobial peptides to combat bacterial infectious agents in bio-films and coatings is still in its early stages, and the research on developing polysaccharides-peptides-based antimicrobial coatings and films is ongoing.”: From the title of the manuscript, “coatings and films” should be removed. The journal to which the authors contributed is ”Coatings. Results and discussion of Coatings and films, even preliminary results, should be included.

(3)   Polysaccharides of various molecular structures exist. What types of actual polysaccharides did the authors use in this paper? There is no description of those polysaccharides at all.

Author Response

Dear Reviewer,

We are grateful for the time and energy you expended on our behalf. Your efforts are most appreciated.

(1)Title, “Proteolysis of β-lactoglobulin assisted by high hydrostatic pressure treatment for development of polysaccharides-peptides based coatings and films”: If the title says "coatings and films," a discussion of coatings and films should be provided in the main text of the authors’ manuscript.

Response: Thank you for this excellent observation and for catching these glaring errors and confusion. We have removed the term polysaccharides-based coating and films and modified it for a better understanding of the reader to avoid any confusion and to present a clear picture of the manuscript. The modified title is “ Proteolysis of β-lactoglobulin assisted by high-pressure treatment releasing peptides and reducing immunoreactivity” and polysaccharides-based coating and film is our ongoing project. We highly appreciated the reviewer's suggestion.

(2) “However, the application of antimicrobial peptides to combat bacterial infectious agents in bio-films and coatings is still in its early stages, and the research on developing polysaccharides-peptides-based antimicrobial coatings and films is ongoing.”: From the title of the manuscript, “coatings and films” should be removed. The journal to which the authors contributed is ”Coatings. Results and discussion of Coatings and films, even preliminary results, should be included

Response: We highly appreciated the reviewer's suggestion. As connected to previous comments, we have changed the title to “ Proteolysis of β-lactoglobulin assisted by high-pressure treatment releasing peptides and reducing immunoreactivity” for better understanding and to avoid any confusion regarding the manuscript.

(3) Polysaccharides of various molecular structures exist. What types of actual polysaccharides did the authors use in this paper?

Response: Thanks for highlighting these points and confusing errors. We are working on the polysaccharides-peptides conjugates-based coating and films by extending the present study which is our present and future project. Briefly, numerous polysaccharides-peptides-based conjugates have been previously published but there were no such reports regarding their coating and films. We are working by extending the present study as an example: getting the benefit of β-lactoglobulin (releasing peptides and reducing immunoreactivity) assisted by high-pressure treatment (present study) and therapeutic, binding enzymes, inhibitor or protein-ligand and drug delivery of application of dextran as cited briefly in T. Mohan, K. S. Kleinschek and R. Kargl, Carbohydrate Polymers, 2022, 280, 118875.

We hope that this improved manuscript will be acceptable for publication in “Coating”. Thank you very much again.